# An Analysis of the Morphology Evolution of YG8 Cemented Carbide by Laser Ablation in the Liquid Phase

**Yujie Fan, Kang Zhao \*, Mengjie Hao, Jing Xia, Xiaoyan Guan and Fanghua Liu**

School of Mechanical Engineering, Jiangsu University of Science and Technology, Zhenjiang 212003, China;
fanyujie@just.edu.cn (Y.F.); haomengjie1029@163.com (M.H.); xiajingdumi@yeah.net (J.X.);
guanxiaoyankb@163.com (X.G.); cylfhua@163.com (F.L.)

**\*** Correspondence: zhaokang20210312@163.com; Tel.: +86-151-6227-8063

**Abstract:** To explore the influence of the number of laser ablations on the shape, geometry, and taper of the pitting structure by laser ablation in the liquid phase, three-dimensional confocal microscopy was used to quantitatively characterize the shape of the surface dotting texture of YG8 cemented carbide and analyze the evolution of the morphology based on the liquid-assisted laser ablation test. The results show that the surface pitting structure of YG8 cemented carbide evolves from a micro-convexity to a crater with the increase in the number of laser ablations, and the bottom of the crater produces a convexity after 7 ablations, the shape of the crater evolves to a trapezoidal shape after 13 ablations, and the shape is stable. The size of the dot texture increases with the number of laser ablations and reaches a maximum value of 396 μm in diameter and 149 μm in depth at the 10th and 12th ablations, respectively. The taper of the dot texture showed a trend of decreasing, increasing, and then decreasing with the increase in the number of laser ablations, and the taper was stable with more than seven ablations. This study lays a theoretical foundation for the control of the dot texture morphology on the surface of YG8 cemented carbide by laser ablation in a liquid-phase environment.

**Keywords:** laser technique; liquid-phase environment; dotting texture shape; taper

## 1. Introduction

Cemented carbide is an alloy material made from hard compounds of refractory metals and bonding metals by powder metallurgy and is widely used in various mechanical engineering equipment and cutting tools. However, high loads and wear are often accompanied during use. Currently, laser ablation of surface microstructures is an effective means to address wear and tear.

Zhang et al. [1] used a high-ultraviolet nanosecond laser to prepare textures on the surface of YG8 cemented carbide, and the optimal texture parameters had a good effect on the friction reduction and wear resistance of the cemented carbide surface. He et al. [2] demonstrated that laser ablation of cemented carbide circular pitted microtextured (CPMT) surfaces resulted in better friction performance than smooth surfaces. Xin et al. [3] found that the microcircular texture of cemented carbide traps impurities and reduces abrasive wear on the cemented carbide surface. Niu et al. [4] searched for the optimal parameters of surface properties by changing the density, diameter, and depth of the weaving area on the surface of medium carbon steel; the best property was obtained for surfaces textured with a depth of 10 μm, area density of 10%, and diameter of 100 μm. Its coefficient of friction was reduced by 38 per cent, and the experiment proved that the depth of the crater is the main factor affecting the friction and wear properties. The surfaces textured optimum depths can effectively prolong the steady state and reduce friction and wear in the variation state, which results in a longer sliding distance in the starved lubrication. Zhang et al. [1] studied the weaving width and area ratio and the influence on the friction reduction and wear properties and found that the optimal parameters of the weaving have a better gain on the friction reduction and wear properties of the surface of the cemented carbide in the

dry friction and solid lubrication states. When the texture width and area ratio parameter combination was 40 μm (45%) and 100 μm (45%), the friction coefficient was reduced from 0.301 to 0.287 and 0.275, respectively. Therefore, it is particularly important to control the size and shape of microtextures. Yang et al. [5] used a conventional two-temperature model (TTM) to optimize laser processing parameters, an optimized hole depth increase of 50 μm, smooth walls, and no residual stresses to improve the quality of turbine blades and blade film cooling holes, thereby improving engine performance. Serguei P. Murzin et al. [6] laser ablated microtextures on the surface of silicon carbide ceramics, which were used to reduce the coefficient of friction of a rolling bearing in operation; the results demonstrate that the laser treatment reduces the coefficient of friction by more than 30 per cent compared to the initial structure. Erkan Öztürk [7] used a laser to ablate microtextures on a tool to enhance machining performances in dry hard turning by decreasing friction between tool and chip, reducing adhesion, reducing cutting forces, and improving wear mechanisms. Aniket Roushan et al. [8] reduced the cutting forces, cutting temperatures, surface roughness, and tool wear by changing the surface morphology parameters of the ablated weave on the tool, which improved machining performance. Above, it can be seen that changing the surface morphology and improving the shape is beneficial in improving the workability and reducing the coefficient of friction. Xin et al. [3] statistically quantified the friction force, coefficient of friction, and wear rate of microcircular weave by the preparation parameters (weave spacing, weave diameter) and geometrical parameters (laser ablation power, ablation speed, number of ablations) and found that the scanning speed, laser power, and the diameter of the microcircular weave have a large effect on the surface properties of the tungsten carbide and that the optimal preparation parameters (laser power: 70 W; scanning times: seven times; scanning speed: 1600 mm/s) can effectively reduce the friction force and the coefficient of friction.

However, most laser processing of carbide surface texture is carried out in air, which is prone to the effect of heat accumulation [9]. Jia et al. [10] studied underwater femtosecond laser ablation of SiC and found that the microporosity depth and the material removal rate increased gradually with the increasing energy of a single pulse and inhibited the occurrence of oxidation reaction, which resulted in a cleaner processed surface. S. vander Linden et al. [11] found that in picosecond-pulsed underwater laser ablation of silicon, the volume of material removed per pulse was better than that in the air environment. Yu et al. [12] found that liquid-phase laser ablation of silicon nitride can effectively inhibit oxidation, but it will reduce the rate of material removal, and with the increase of scanning speed, the surface roughness increases, and the depth of laser ablation decreases. Zhou et al. [13] utilized water-assisted laser ablation in the different water layers and found that the liquid-phase ablation environment can get clearer micrographs and smaller heat-affected zones, the liquid-phase environment makes the inner wall of the grooves clean and smooth, and the depth of the grooves decreases with the increase of the liquid-phase flow rate. PANG Minghua et al. [14] processed YT15 cemented carbide in a liquid-phase environment and found that the micro-pit distribution on the surface of liquid-phase assisted laser processing is more uniform (with basically no oxidation and recasting of molten material), the assisting liquid plays a cooling and protecting role on the laser processing surface, and the surface contact angle is minimized (55°). In addition, the friction coefficient of the surface texture processed by liquid-phase-assisted laser processing is the smallest, and the wear amount is improved to a certain extent. Yang et al. [4] found that the quality of laser microtextures is very important for aerospace turbine blades and blade film cooling holes, in which the diameter–depth variation of ablation is related to the effective surface area of ablation, laser intensity, ablation time, number of pulses, the energy of a single pulse, and the target material of ablation. Zhu et al. [15] found that the pit taper is related to the angle of laser ablation and the number of ablation times, and the edge quality of microtextured structures can be effectively improved by the method of laser compensation.

Existing research has demonstrated that liquid-phase laser ablation produces good morphology, lessens the formation of recast layers and cracks, and enhances the heat-

building effect during laser ablation and the recasting phenomenon during processing. Nevertheless, the influence of liquid-phase-assisted laser ablation on YG8 cemented carbide has not been reported. To investigate the impact of ablation times on the morphology of liquid-phase laser ablation of YG8 cemented carbide, this study uses a mixed liquid of anhydrous ethanol and hydrogen peroxide as a propellant to facilitate laser ablation processing of YG8 cemented carbide. There is a significant improvement in the ablation morphology compared with other machining processes [1], and it is particularly outstanding for the reduction of heat-affected zones, recast layers, and cracks [16], which provides a theoretical basis for the machining and shape control of the point weave structure on the surface of YG8 cemented carbide. This research has some guiding significance for the use of laser ablation to achieve high-quality shapes, sizes, and taper controls in key industrial areas such as in the machining and manufacturing of micro-tools [7], engine fuel injectors, and cooling holes in turbine blades [5].

## 2. Materials and Methods

### 2.1. Test Material

The test material was YG8 cemented carbide, with a size of 20 mm × 20 mm × 5 mm. Sandpapers with 800 mesh, 1000 mesh, and 2000 mesh were used to polish the surface, and then the surface was polished to the roughness Ra < 0.8 μm, and finally cleaned with deionized water and acetone, and then left to dry naturally in the air for spare parts.

### 2.2. Test Equipment

The laser system was a Nd: YAG laser with a wavelength of 1064 nm and a pulse width of 15 ns. The laser energy density was 200 J/mm$^2$, the pulse frequency was 20 KHZ, the spot diameter was 350 μm, and the number of ablations ranged from 1 to 14.

The auxiliary liquid is a mixture of hydrogen peroxide and anhydrous ethanol in the ratio of 30:1 by volume, controlled by volumetric measurement of the workpiece at a height of 1 mm above the liquid surface. Figure 1 is the schematic diagram of the liquid-phase laser processing process, and the surface dot-woven array is shown in Figure 1. In total, ten sets of experiments were performed in which the room temperature was controlled to be constant at 20° to ensure the reproducibility and robustness of the observed effects.

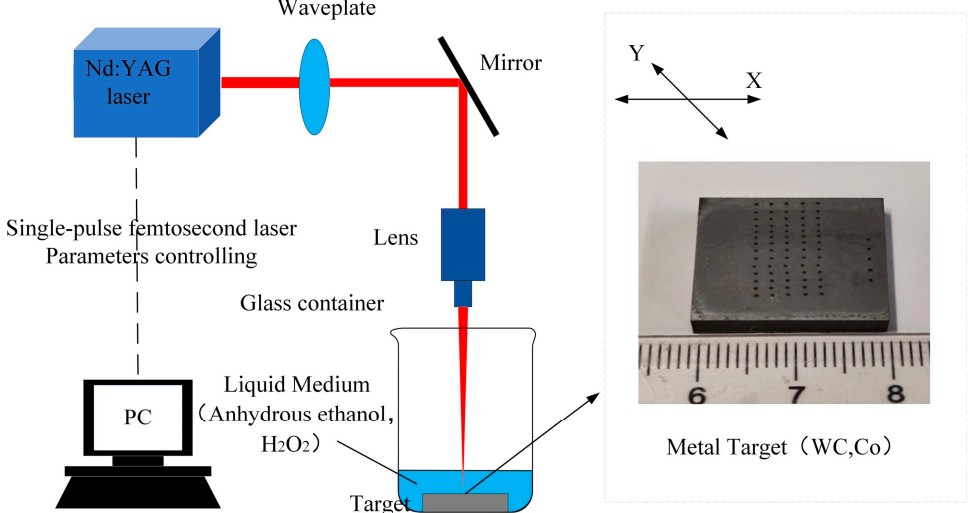

**Figure 1.** Schematic diagram of laser processing technology in liquid phase.

## 3. Results and Discussion

### 3.1. Analysis of the Shape Evolution of the Laser Ablation Dot Texture in a Liquid-Phase Environment

(1)   Three-dimensional morphology of laser ablation dot texture in liquid-phase environment

Figure 2 shows the three-dimensional morphology of the dot texture of the YG8 cemented carbide surface ablated by laser in a liquid-phase environment. It is evident that as the number of ablations increases, more material is gradually removed from the ablated surface, the center region of the dot texture changes from a slight bump to a pit, and the pit's depth increases gradually. In contrast, the edge region of the bump structure's dot texture does not significantly change as the number of ablations increases and produces very little recast melt.

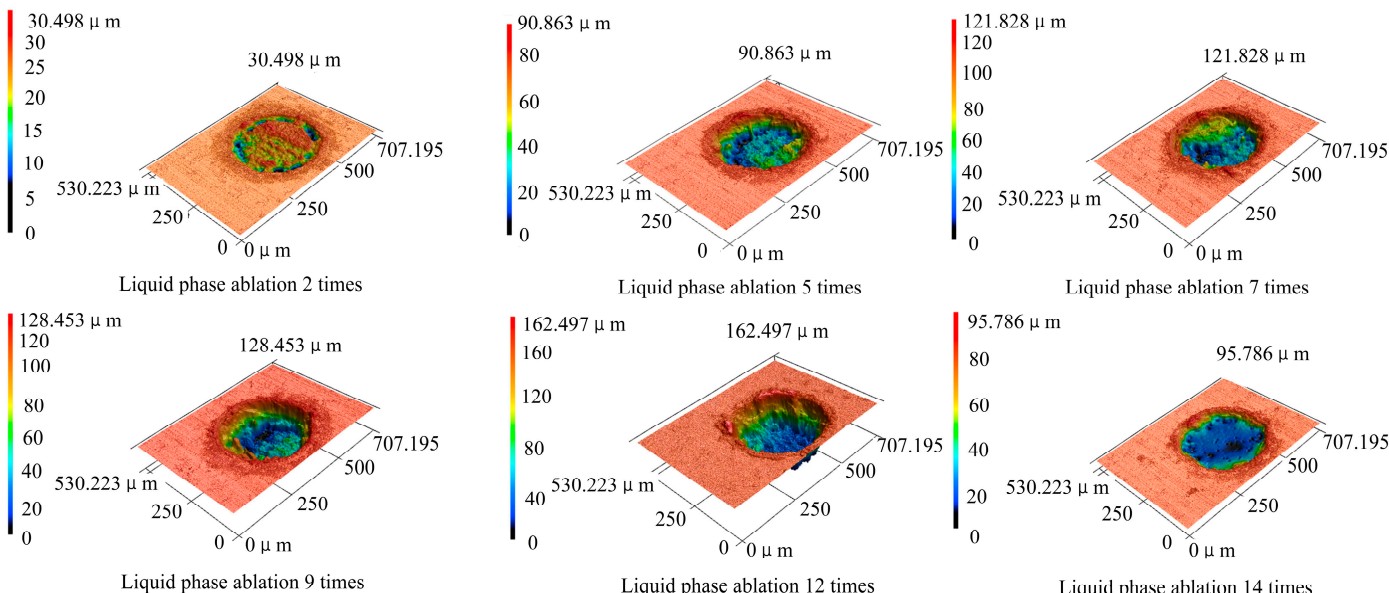

**Figure 2.** Three-dimensional morphology of dot texture of YG8 cemented carbide surface by laser ablation in liquid phase.

(2)   Laser ablation of dot texture two-dimensional profiles in a liquid-phase environment

Figure 3 shows the two-dimensional contours of the dot texturing centers for different numbers of liquid-phase ablations. It can be seen that when the number of ablation is one, there is a bump around the dot texture, and the ablation shape of the substrate surface is a crater at this time. This is because the substrate surface will create a local melt pool during one ablation. The melt will then flow outward from the melt pool's center under the recoil pressure, creating a micro-pit on the substrate surface and a pit surrounding the micro-bumps [17]. The ablated surface of the substrate is convex when the number of ablations is two. This is because as the number of ablations increases, remelting takes place in the ablated area of the substrate, the temperature in the melt pool's center is at its highest, and surface tension causes a strong inward Marangoni flow [18], which causes the melt to flow to the melt pool's center and eventually form a convex shape.

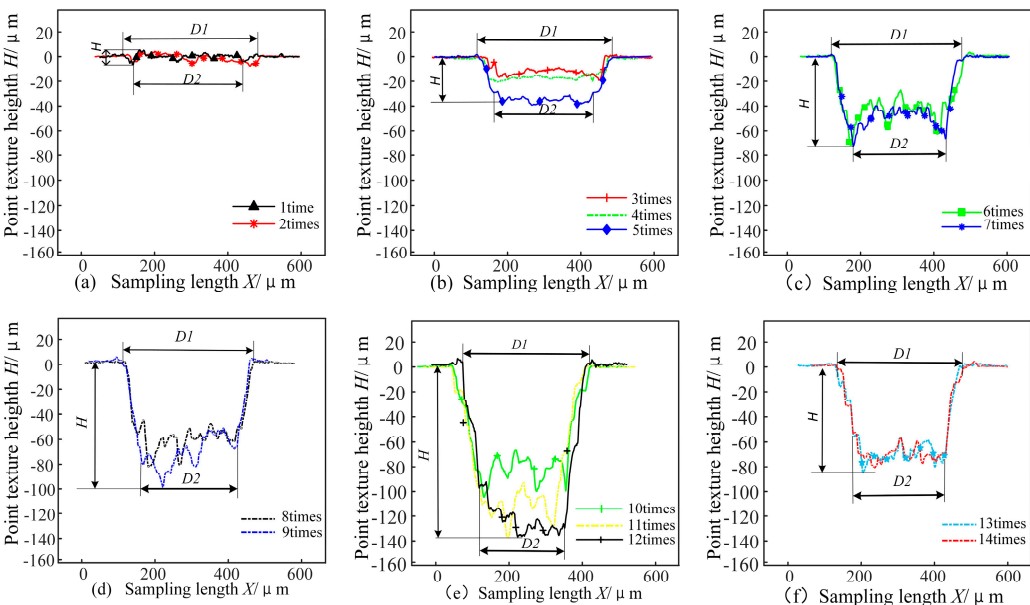

**Figure 3.** Two-dimensional profile of the center diameter of dot texture under different ablation times.

When there are more than three ablations, a pit gradually forms in the center. This is due to the increase in the number of ablations, resulting in the accumulation of heat, and because the center part of the bulge is subjected to the most heat, part of the material reaches the ablation threshold, and the material removal phenomenon takes place [19]. When the number of ablations reaches five, rapid ablation begins. At this point, the energy in the center of the protruding part of the melt gradually starts to accumulate, and the liquid acts as a propellant, which gradually forms a pit with depth, and the depth of the pit is more uniformly ablated. However, due to the recoil pressure, the melt at the bottom of the pit will be extruded, and the liquid will gradually take away most of the removed material and melt, thus reducing the accumulation of material [20]. At 9–12 times into the steady ablation stage, both the ablation depth and recoil pressure increase, resulting in smoother ablation [21].

When there are 12 to 14 laser ablations, the depth of the pit decreases with a small convex shape around it. During this period, the bottom of the pit produces a plasma shielding effect [22], and with the increase of the depth of the liquid through the laser, loss increases, which in turn weakens the laser ablation effect on the pit, resulting in a decrease in the depth of the ablation pit. As the substrate absorbs a large amount of heat in a short period, the recoil pressure formed by the material in the center of the melt pool is too large, which causes the liquid bubbles to explode, leading to the intensification of the flow of the melt at the bottom of the pit and spreading to the surrounding area of the pit and forming a part of the molten material around the pit.

(3) Dimensional changes of laser ablation dot texture in liquid-phase environments

Figure 4 shows the graphs of the variation of the weaving height and the top and bottom diameters of the laser-ablated spots with the number of ablations in the liquid-phase environment. It can be seen that when the laser ablates 1–2 times, the diameter and depth of the top of the dot texture decreases with the increase of laser energy, and the diameter and depth of the top of the crater are up to 330 μm and 6 μm, respectively, and the ablation efficiency is lower than that of the thermal expansion efficiency at this time, and thus the micro-bump is generated. At 3–5 times ablation, the diameter and depth of the texture increase with the rise of laser energy. This happens because of the increase of heat absorption in the center part of the micro-bump material and liquid, resulting in the reduction of the material. After rapid ablation, a pit with a larger diameter and depth is formed. The top of the pit reaches a diameter of 380 μm and a depth of 39 μm

after the 5th ablation. At 5–8 ablations, the top and bottom diameters of the dot texture show a decreasing trend, while the depth ablation of the dot-textured changes faster due to the creation of a bulge at the bottom of the pit, which in turn reduces the ablation of the bottom edge of the pit. At 8–10 ablations, the diameter and depth of the dot texture increase rapidly, the ablation of the bottom edge of the pit increases, and the efficiency of the liquid-propelled ablation is greater than the energy loss, with a maximum diameter and depth of 396 μm and 114 μm of the dot texture, respectively. At 11–12 times, the diameter of the dot texture decreases with the increase of the depth of the dot texture because the ablation center is shifted to the center of the dot texture, and the maximum depth of the dot texture is 149 μm. Finally, at the 13th to 14th times of ablation, the diameter and depth of the dot texture change slowly with the increase of the laser energy density, and the diameter and depth of the dot texture stabilize at around 353 μm and 85 μm, respectively. As the number of laser times increases, the energy generated also accumulates. This leads to a greater loss of energy through liquid ablation, and the substrate material is continuously heated by the laser to form a plasma [23]. The plasma, with the increase in the laser energy density of the laser beam formation, has a certain shielding effect, so the ablation depth decreases and then tends to be stabilized.

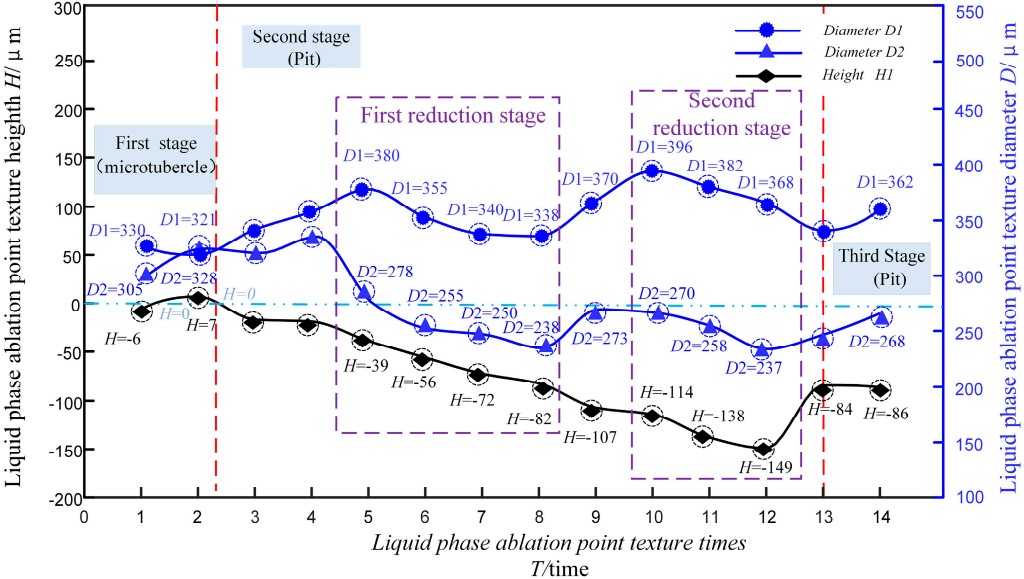

**Figure 4.** Changes in the height and diameter of the dot texture with the number of ablations in the liquid phase.

(4)   Modeling the shape evolution of laser ablation dot texture in a liquid-phase environment

Figure 5 shows the shape evolution model of the YG8 cemented carbide dot structure with the number of liquid-phase laser ablation. It can be divided into six stages. Stage 1: evaporation stage. The number of ablations is 1–2, and the workpiece produces a small number of pits and bumps. This stage of ablation does not produce much energy, the impact on the workpiece is limited, and the surface of the workpiece removes fewer substances. And then, as the surface of the YG8 reaches the boiling point and forms upward metal vapor, the liquid surface begins to produce an evaporation phenomenon [24]. Stage 2 is the molten pool stage. The number of ablations is 3–5, the surface material melts and forms a molten pool, and the workpiece is also gradually subjected to thermal expansion and radiates around [25]. Stage 3: the molten pool vortex. The number of ablations is 6–7, with the increase in the number of ablations producing a larger recoil pressure exacerbating the depression of the molten pool, in which double vortex is obvious; it is caused by the natural convection in the molten pool and the Marangoni convection [26].

The natural convection is driven by buoyancy forces of the density of the liquid metal affected by temperature. It causes the hot melt to float. Marangoni convection occurs when the surface tension of a liquid metal is affected by a temperature gradient, causing the high-temperature melt to flow towards the low-temperature region. The combined effect creates a center–edge–bottom–center vortex path in the melt pool. According to the conservation of momentum, the high-temperature melt flows towards the low-temperature region and exerts a downward recoil pressure on the melt pool. Stage 4: high-pressure melt pool. The number of ablations is 8–9. The surface of the melt pool shows high pressure and high velocity, while the pressure and velocity inside the melt pool are low, so the heat is transferred to the inside, and the vortices converge toward the center of the melt pool as the temperature and velocity decreases, generating upward velocity vectors at the surface of the melt pool, the downward trend is suppressed [27]. However, the upward flow rate on both sides of the melt pool is very high [28], and the pressure inside the melt pool is very high, which remains downward ablated by surface tension and buoyancy. Stage 5: the melt pool recoil. The number of ablations is 10–12. With the increase of ablation depth, the vortex inside the melt pool weakens, which inhibits the surface depression; this stage of the melt pool surface produces upward velocity vectors, which inhibits the downward tendency to reach its peak. Stage 6: shielding and stacking. The number of ablations is 13–14. The melt pool is kept oscillating under recoil pressure, surface tension, buoyancy, and gravity, and the plasma produced by its ablation produces a shielding effect on the laser ablation [29], resulting in the buildup of the molten cladding, and the laser ablation has no significant change on the melt pool, thus achieving dynamic equilibrium.

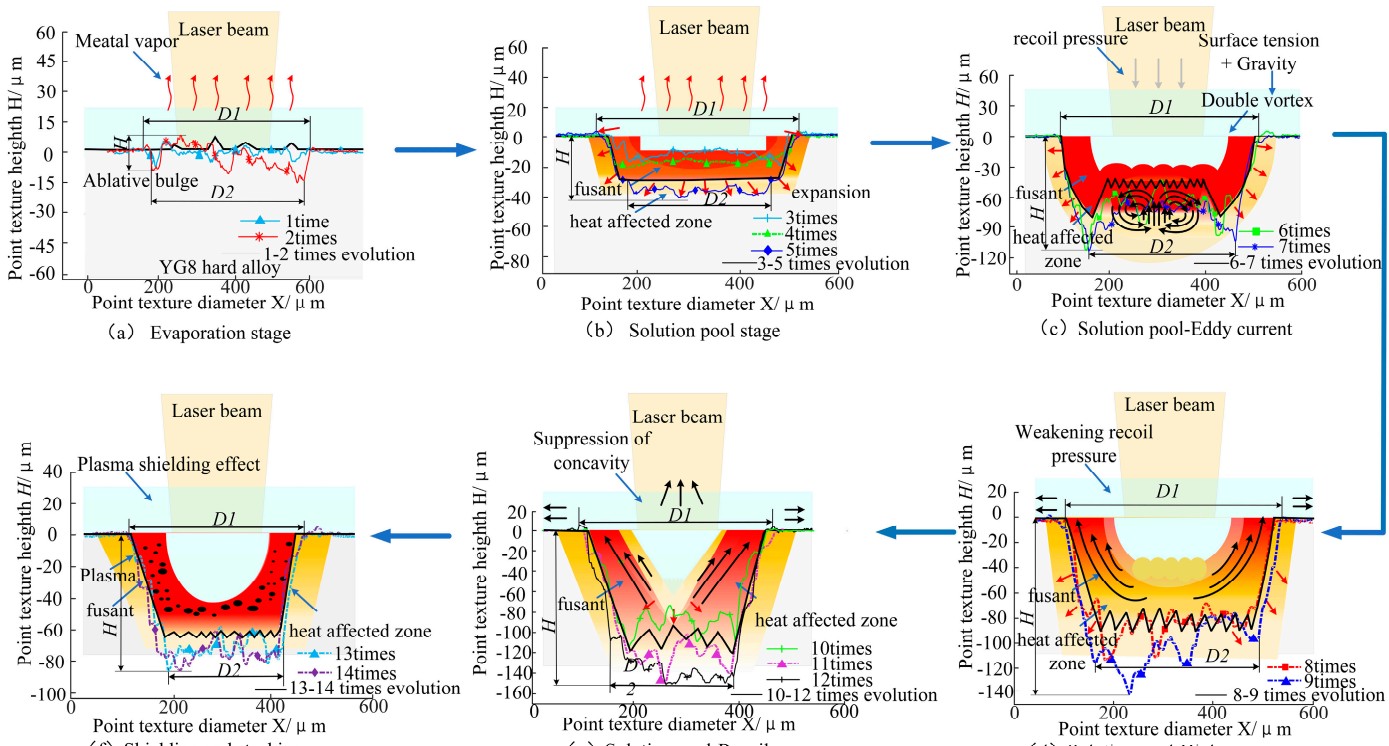

**Figure 5.** Shape evolution model of dot texture induced by laser ablation in liquid-phase environment.

### 3.2. Analysis of the Taper of Ablation Dot Texture Craters in Liquid-Phase Environments

Figure 6 shows a schematic diagram of the dot texturing taper of the liquid-phase laser ablation, whose taper $V$ is calculated by the following formula:

$$V = \frac{D1 - D2}{H} \times 100\% \tag{1}$$

where the depth of the hole cross-section is $H$, and $D1$ and $D2$ are the diameters of the top and bottom surfaces of the pit, respectively [30].

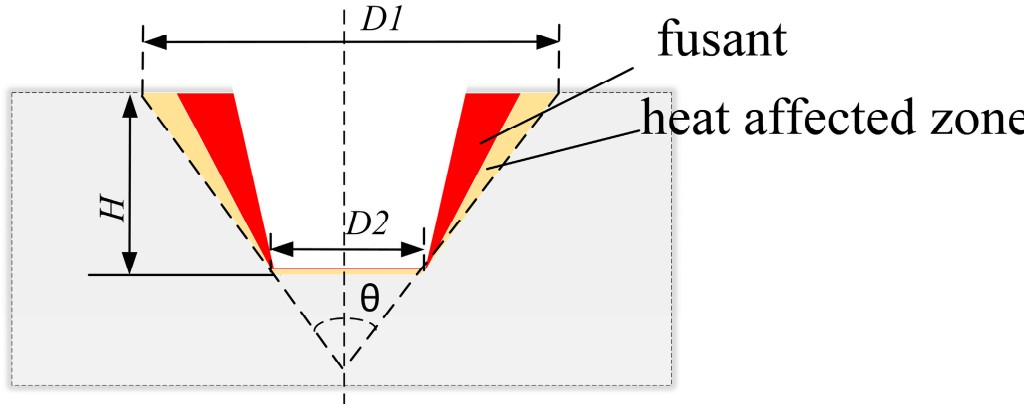

**Figure 6.** Schematic diagram of the taper of the dot texture.

As shown in Figure 5, after two ablations, the taper of the dot texture is 4.17, and the number of laser ablations in the initial stage is less than after crater removal, with the hole taper being larger. The taper of 5 times ablation is 2.62; as the number of ablations increases, the pit removal increases, and the hole taper decreases gradually. The taper of 7 times ablation is 1.25, which enters into the stage of rapid ablation, and the center of the pit ablation is gathered in the middle; as shown in Figure 5d–f, the rate of laser ablation slows down gradually in this stage, and the depth of ablation tends to be stabilized due to plasma shielding effect, the taper is stabilized at about one.

Figure 7 shows the graph of the taper variation of laser-ablated dot texture with the number of ablations in the liquid-phase environment, and it can be seen that the taper shows a decreasing, then increasing, and then decreasing trend with the increase of the number of ablations. The taper trend is divided into three stages [31]: The first stage is unstable because the initial ablation number is small (1~2), and the change in diameter difference between the top and bottom of the dot texture will increase, while the depth change is small, and the taper fluctuation range is large. The second stage is the stabilization stage when the number of ablations is 3–4, the ablation rate is accelerated, the diameter and depth of the pit top and bottom increase simultaneously, and the calculated value is smaller according to the taper equation. At the fifth ablation, the diameter of the top of the pit increases while the bottom decreases, and the difference between the diameters of the top and bottom increases more rapidly than the depth, resulting in taper increases to 2.62. At the sixth ablation, the top and bottom diameters of the dot texture decrease while the depth gradually increases, so the taper decreases. The third stage is 7–14 times of stabilization; the difference between the diameters of the top and bottom of the crater and the ablation depth have the same trend, so the taper fluctuates in a certain range.

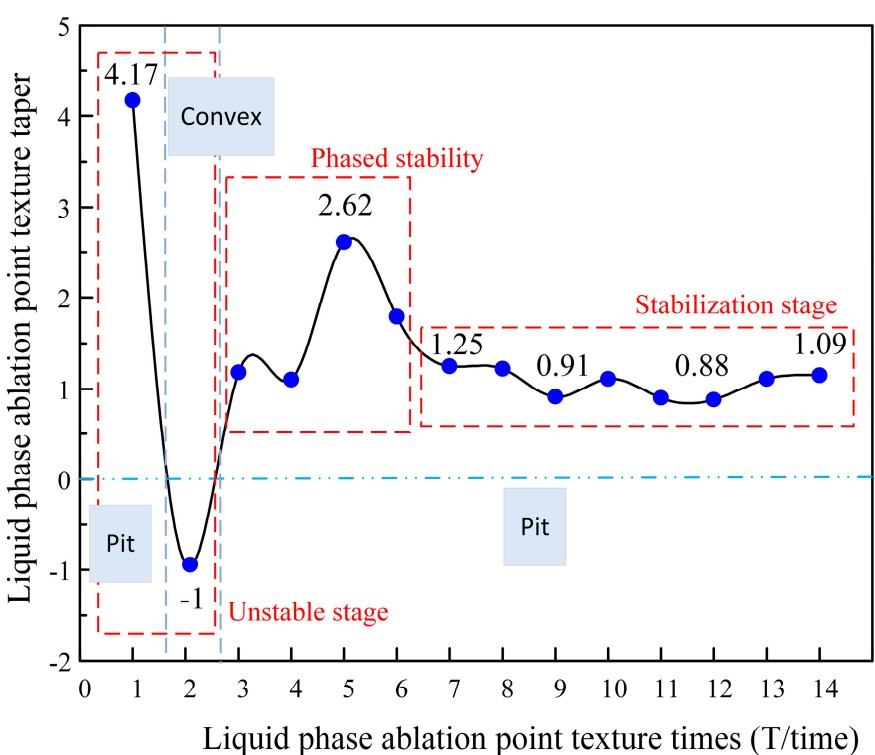

**Figure 7.** Change of taper of dot texture with the number of ablations in the liquid phase.

## 4. Conclusions

In this study, the liquid-phase laser ablation technique was used to investigate the effect pattern of the number of laser ablations on the liquid-phase laser ablation of YG8 cemented carbide. There are three findings as follows.

(1) Laser ablation of YG8 cemented carbide morphology was divided into six stages: The first stage was the evaporation stage, where the shape of the convex and small size changed. The second stage was the melt pool stage, the heat-affected zone began to expand, and the shape of the ablation was a trapezoidal shape, the more regular shape. The third stage was the melt pool vortex: the bottom of the pit produced a rise and characteristics of a pressure vortex. The fourth stage was the high-pressure melt pool: the bottom of the pit rise disappeared, and with it, the pressure vortex disappeared, and the shape took on an irregular trapezoidal shape. The fifth stage was the molten pool recoil: The shape evolved into a trapezoidal shape with a smaller taper, and the ablation depth was suppressed. And the final stage was shielding and stacking: the pits' shape changed to a more regular trapezoidal form with a smaller taper, and the shape was stabilized due to the plasma shielding effect.

(2) The number of laser ablations had a significant effect on the geometry of the surface texture. When the number of laser ablations was varied from 1 to 4 and 8 to 10, the diameter and depth of the top and bottom of the dot texture showed an increasing trend. The number of ablations was 5–8 and 10–12, the diameter of the dot texture decreased, and the depth increased. After more than 13 ablations, the diameter of the texture pits increased, the depth decreased, and the dot texture size changed slowly.

(3) The number of laser ablations had a great influence on the taper of the pits, with an overall trend of decreasing, increasing, and then decreasing. When the number of laser ablations was 3–4, the taper of the pits was small; when the number of ablations was more than 7, the taper of the pits tended to stabilize gradually.

This study reveals the effect and mechanism of the number of liquid-phase laser ablation on the surface dot texture morphological changes of YG8 cemented carbide and achieves the control of liquid-phase laser ablation dot texture geometry and size. Since the methodology study was limited to the morphological changes induced by the specific

workpiece type and the number of laser ablations, parameters such as laser power density, pulse width, etc., are a step toward unexplored machining properties in future studies. This research is expected to be applied to rolling bearings, cutting tools, aero engine blades, and other transmission parts.

**Author Contributions:** Conceptualization and validation, Y.F.; writing—original draft preparation and visualization, K.Z.; formal analysis and methodology, M.H.; resources, J.X.; supervision, X.G.; investigation, F.L.; writing—review and editing, Y.F. All authors have read and agreed to the published version of the manuscript.

**Funding:** This research was funded by the National Natural Science Foundation of China, grant number 12202167.

**Institutional Review Board Statement:** Not applicable.

**Informed Consent Statement:** Not applicable.

**Data Availability Statement:** Data are contained within the article.

**Acknowledgments:** The authors are grateful to the School of Mechanical Engineering, Jiangsu University of Science and Technology, for providing equipment support.

**Conflicts of Interest:** The authors declare no conflict of interest.

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
