# Peer review of "An Analysis of the Morphology Evolution of YG8 Cemented Carbide by Laser Ablation in the Liquid Phase"

_coatings, doi:10.3390/coatings13122061_

Round 1

Reviewer 1 Report

Comments and Suggestions for Authors

The article focused on exploration of the influence of laser ablations on the surface texture, shape evolution, and taper of the pitting structure that is interesting and relevant for material processing applications. However, there are a few points that could be addressed for clarity and further elaboration:

1-Have these findings been replicated or validated through multiple experiments or by other research groups? Ensuring the reproducibility and robustness of the observed effects would strengthen the study's credibility

2-Authors mentioned potential applications in rolling bearings, cutting tools, aero-engine blades, and transmission parts. Please provide more insights into how controlling the dot texture's geometry and size through laser ablation, specifically benefit these industries? Are there any performance enhancements or functional improvements expected by altering the surface morphology in these applications?

3-The relationship between the number of ablations and the geometry of the surface texture is well-documented. However, it might be beneficial to elaborate on the precise mechanisms driving these changes. What physical phenomena or material properties contribute to the observed variations in diameter, depth, and taper of the dot texture?

4-While the stability of the crater shape and taper after a certain number of ablations is noted, could you discuss the practical implications of these findings? How might this stability influence or guide industrial processes involving YG8 cemented carbide surfaces?

Comments on the Quality of English Language

  1. "the emergence of the pressure vortex," defining the nature or characteristics of the "pressure vortex" could provide readers with a clearer understanding.

  2.  "the shape of the pits took on a more regular trapezoidal shape" could be revised for better clarity and conciseness.

Reviewer 2 Report

Comments and Suggestions for Authors

Journal: Coatings

Manuscript ID: coatings-2756381

Title: Analysis of Morphology Evolution of YG8 Cemented Carbide by Laser

Ablation in Liquid Phase

Authors: Fan Yujie, Zhao Kang *, Hao Mengjie, Xia Jing, Guan Xiaoyan, Liu

Fanghua

The paper investigates the effect of laser ablation on WC surface morphology. The authors investigated the depth of pits, diameter, taper, and proposed an interesting ablation scheme. The topic is quite interesting, in line with the journal's subject matter, and the results are quite new. However, I would say that the volume of the paper itself (not in terms of the number of pages, but in essence) is not very high for the journal level. As it seems to me, the task set is rather small. At the same time, the work is well done and of high quality, so there are few specific comments:

1.                  The quality of images needs to be improved. In the submitted manuscript, images (particularly graphics) have low resolution and jpeg artifacts. This may be a consequence of pdf compression, so should be checked.

2.                  The pulse widths are specified, but the intervals between them are not. I suppose with other parameters the pattern may change significantly.

3.                  Need to specify the authors' affiliation.

4.                  The most important flaw in the paper: the introduction discusses that laser ablation is used to reduce wear. However, this is not investigated or discussed further in the paper in any way. Experimental results should be discussed and conclusions should be drawn - at what stage of ablation it is better to stop, how the wear will change.

Reviewer 3 Report

Comments and Suggestions for Authors

1. What is the main question addressed by the research?

The main question addressed by the research is how the number of laser ablations influences the shape, geometry, and taper of the pitting structure on the surface of YG8 cemented carbide during laser ablation in a liquid phase.

2. Do you consider the topic original or relevant in the field? Does it address a specific gap in the field?

The topic appears to be relevant in the field, particularly in the realm of laser ablation techniques and their impact on material surfaces. The research aims to explore the influence of the number of laser ablations on the morphology of YG8 cemented carbide, a material widely used in mechanical engineering equipment and cutting tools. The study addresses the specific gap of understanding how liquid-phase laser ablation affects the surface dot texture of YG8 cemented carbide, providing insights into the control of morphology in a specific environment.

3. What does it add to the subject area compared with other published material?

The research adds to the subject area by investigating the specific effects of liquid-phase laser ablation on the surface dot texture of YG8 cemented carbide. While laser ablation techniques have been explored in various studies, this research delves into a previously unreported aspect—the impact on YG8 cemented carbide in a liquid-phase environment. By analyzing the morphology evolution, the study contributes valuable insights into the shaping and control of dot texture on this particular material, potentially opening avenues for applications in areas such as rolling bearings, cutting tools, and aero-engine blades.

4. What specific improvements should the authors consider regarding the methodology? What further controls should be considered?

The authors could consider a few improvements and additional controls for their methodology:

4.1-Controlled Variables: Ensure precise control over variables that might affect the results, such as ambient conditions, material properties, and the stability of the laser system.

4.2-Replication: Conducting experiments with a higher number of replications would enhance the reliability of the findings.

4.3-Validation: Consider validation techniques or comparisons with alternative methodologies to strengthen the robustness of the results.

4.4-External Factors: Discuss potential external factors that could influence the outcomes and address how they were mitigated or controlled.

4.5-Statistical Analysis: If not already done, consider incorporating statistical analysis to quantify the significance of observed trends and variations.

4.6-Discussion of Limitations: Clearly articulate the limitations of the methodology and potential sources of bias to provide a comprehensive understanding of the study's scope.

These suggestions aim to enhance the methodological rigor and contribute to the overall validity of the research.

5. Are the conclusions consistent with the evidence and arguments presented and do they address the main question posed?

Yes, the conclusions drawn in the passage appear to be consistent with the evidence and arguments presented. The study outlines a comprehensive analysis of the morphological evolution of YG8 cemented carbide through liquid-phase laser ablation. The conclusions address the main question posed in the research, summarizing the observed changes in surface dot texture morphology, the influence of the number of laser ablations, and the taper of the dot texture.

The study effectively connects the experimental results with the research question, providing insights into the various stages of the laser ablation process and their impact on the material. The conclusions are derived from the data presented throughout the passage, offering a clear and logical interpretation of the findings.

6. Are the references appropriate?

Almost

7. Please include any additional comments on the tables and figures.

The manuscript has a potential originality and show interesting results and discussion points.
